# Effects of Perinatal Antibiotic Exposure and Neonatal Gut Microbiota

**DOI:** 10.3390/antibiotics12020258

**Published:** 2023-01-28

**Authors:** Chiara Morreale, Cristina Giaroni, Andreina Baj, Laura Folgori, Lucia Barcellini, Amraj Dhami, Massimo Agosti, Ilia Bresesti

**Affiliations:** 1Department of Medicine and Surgery, University of Insubria, Via JH Dunant 5, 21100 Varese, Italy; 2Department of Paediatrics, Children’s Hospital V. Buzzi, University of Milan, 20133 Milan, Italy; 3Newborn Services, John Radcliffe Hospital, Oxford University Hospital, Oxford OX3 9DU, UK; 4Department of Woman and Child, “F. Del Ponte” Hospital, ASST-Settelaghi, 21100 Varese, Italy

**Keywords:** antibiotic, neonate, neonatal intensive care unit, preterm, probiotics, gut microbiota, microbiome

## Abstract

Antibiotic therapy is one of the most important strategies to treat bacterial infections. The overuse of antibiotics, especially in the perinatal period, is associated with long-lasting negative consequences such as the spread of antibiotic resistance and alterations in the composition and function of the gut microbiota, both of which negatively affect human health. In this review, we summarize recent evidence about the influence of antibiotic treatment on the neonatal gut microbiota and the subsequent negative effects on the health of the infant. We also analyze the possible microbiome-based approaches for the re-establishment of healthy microbiota in neonates.

## 1. Introduction

Antibiotic therapy is fundamental to the treatment of infectious diseases in humans and represents one of the greatest medical advances of the last century [1]. However, antibiotic overuse is associated with important negative consequences, such as the spread of antibiotic resistance. Despite this, the worldwide use of antibiotics is increasing [2]. Furthermore, in recent years, several studies have demonstrated that although most antibiotic courses are associated with limited adverse effects, antibiotics can negatively affect the gut microbiota, with short- and long-term consequences on the host’s health [3]. Such alterations may be particularly significant in the first years of life, which represent critical windows of growth and development for both the host and the gut microbiota [4,5]. In this context, the use of antibiotics in the perinatal period, either in the mother or in the neonate, may have dramatic and long-lasting consequences for the infant’s gut microbiota. Antibiotic-induced changes in the infant’s saprophytic microbial community have been associated with intestinal diseases and a predisposition to a range of illnesses later in life [6,7,8]. This review synthesizes the more recent evidence from the literature suggesting that perinatal antibiotic treatment may influence the homeostasis and function of the neonate’s gut microbiota, with negative consequences on the infant’s health, including the early onset of antibiotic resistance. The main evidence pointing to microbiome-based approaches for the re-establishment of a healthy microbiota in neonates is also discussed.

## 2. Gut Microbiota Composition in the Early Stages of Life

The enteric microbiota is composed of a rich and dynamic population of microorganisms, including bacteria, viruses, archaea, fungi, and protozoa. The predominant microbial community is that of bacteria, consisting of approximately 3.8 × 10^13^ cells, with a number of genes exceeding 100–150-fold the number of human genes [9,10]. The gut microbiota represents the most abundant microbial population in the human body, playing a key role in the control of host health, both locally and systemically. In the gut, the saprophytic commensal bacteria influence nutrient and drug metabolism, the development of the immune system, and the defense against pathogenic microorganisms [11]. This rich ecosystem is not static, and undergoes significant variations during the host’s lifespan, especially in extreme conditions, i.e., during infancy and aging, when the microbiome has different degrees of diversity and is composed of different representative taxa with respect to the healthy adult gut microbiome [12]. Such characteristic variations of the gut microbiome composition may not only depend upon external influences but also on the host’s health status [12,13]. In this context, the period from conception to the age of 2 years represents a critical growth window, characterized by the rapid development of metabolic, endocrine, neural, and immune functions, the correct programming of which may reduce the risk of developing diseases later in life [5]. Fetal exposure to microbial metabolites, as well as microbial colonization of the neonatal gut, helps underlay the formation of an interconnected network between the host and the symbiotic microbial community. This network is important for participating in the maintenance of the gut’s barrier function, influencing immune responses and nutrient absorption and promoting optimal growth and neurodevelopment [4,11]. In the early stages of life, the gut microbiome becomes more diverse, reaching a stable adult-like composition by 2–4 years of age [12]. Any factor that alters the composition or function of the gut microbiota (i.e., dysbiosis) in this early period of life may represent a risk factor, predisposing the host to diseases such as allergies, asthma, cardiovascular disease, obesity, inflammatory bowel disease, irritable bowel syndrome, and neurodevelopmental disorders [4,13]. Several factors may influence the composition and activity of the infant gut microbiota, such as gestational age, mode of birth, infant feeding, environment, hospitalization, and antibiotic treatment [13]. In order to evaluate the consequences of these perinatal exposures on the shaping of the infant microbiota, it is crucial to define the characteristics of a normal infant microbiome. In the next paragraphs, an overview of the available data on the composition of the healthy gut microbiome in neonates is presented, distinguishing between prenatal and postnatal influences.

### 2.1. Prenatal Period

Various studies have shown that the amniotic membrane, amniotic fluid, placenta, meconium, and umbilical cord blood harbor different microbial communities in healthy pregnancies, contravening the conventional concept that the fetus is sterile [14]. Interestingly, the levels of placental, amniotic, and meconium microbiota are more elevated in preterm infants, suggesting a potential influence of prenatal microbial exposure on the growth and length of gestation [15]. However, it is not yet clear-cut whether this microbial presence is a cause or effect of preterm birth and impaired growth. The in utero microbial presence is, however, much criticized, owing to possible sample contamination, limited detection approaches, and evidence of bacterial viability [16]. Indeed, intrauterine life is characterized by limited exposure to microbes, and neonatal gut colonization occurs prevalently during and after birth [12,14].

An important issue to be considered is the possible in utero exposure to microbial metabolites produced by the mother’s gut microbiome, which may influence fetal growth and development, contributing to early signaling pathways at the fetal–maternal interface [4]. Notably, some metabolic alterations occurring during pregnancy may be correlated with changes in gut microbiota. Whether such changes may influence the developing fetus, however, remains to be clarified [4].

### 2.2. Postnatal Period

The colonization of an infant’s gut is fundamental for the development and maturation of the immune function and, consequently, of the health of the individual [17]. During the postnatal period, the shaping of the microbiome composition occurs in two major colonization steps, interleaved by the weaning period at about 6 months of age [12]. Following birth, the neonatal gut is rapidly colonized by facultative anaerobes, with a relatively high abundance of strains belonging to the families of Enterobacteriaceae, Bifidobacteriaceae, and Clostridiaceae, and low levels of Lachnospiraceae and Ruminococcaceae [18,19,20]. These microorganisms, by consuming the available oxygen, generate an appropriate anaerobic atmosphere, allowing the successive growth of strict anaerobic taxa with an increase in the overall bacterial diversity. The establishment of an adult-like microbiota, with a high relative abundance of Bacteroidetes and Firmicutes, is observed by 1–3 years of life, after weaning, when the food source and composition change from being liquid and rich in fat to being solid and rich in carbohydrates [20,21]. The composition and function of the microbiota still develop after weaning and remain significantly different from the healthy adult microbiota until 7–12 years of age to support the ongoing development of the individual [22].

The infant microbiota is susceptible to a variety of environmental factors, including the mode of birth, prematurity, the birth location (home versus hospital), diet (for example, breastfeeding versus formula feeding), maternal gestational diet and weight, pet ownership, disease state and stress, and antibiotic treatment, with long-lasting consequences on the physical and mental health of the individual, at least for some of these effects [12,23].

The mode of delivery represents one of the first factors influencing the neonatal gut microbiota, although the effects of the mode of birth on microbiota composition are no longer apparent by 6–8 weeks of age [18] or, according to other studies, by the first 2 years of life [24]. Vaginally delivered term infants are initially colonized by a microbiota that belongs to the maternal vaginal flora, especially *Lactobacillus*, *Prevotella*, and *Sneathia* spp. [14,25]. Cesarean-section-born term infants have a gut microbiota that resembles the maternal skin, with high levels of *Clostridium difficile*, *Streptococcus*, *Staphylococcus*, and *Propionibacterium* spp., and a reduced concentration of *Actinobacteria* and *Bacteroidetes* [14,25]. However, in one study, no correlation was observed between the specific maternal skin microbiota and the microbiota of their cesarean-section-born infants, suggesting the existence of other nonmaternal sources for bacterial colonization [26].

Another important factor influencing postnatal microbiota composition is the gestational age at birth. The microbiota of preterm infants’ (gestational age < 34 weeks) is characterized by low diversity and increased levels of potentially pathogenic bacteria [14], even if interindividual variations remain elevated. It has been suggested that a preterm infants’ microbiota can be distributed into five or six common bacterial clusters, defining a “preterm gut community type”, each one characterized by a specific genus of dominance. However, the establishment of the gut microbiota in preterm neonates still needs to be fully clarified [14]. For preterm infants, the environment has a more influential role, and their microbiota may be composed of bacterial species which belong to hospital surfaces and feeding and intubation tubing, such as *Staphylococcus epidermidis*, *Klebsiella pneumoniae*, *Escherichia coli*, *Enterobacteria*, and *Streptococci*; subsequently, these are replaced by other anaerobic bacteria (such as *Bifidobacterium*, *Bacteroides*, *Clostridium*, and *Eubacterium*) during the end of the first week of life [14]. *Bifidobacteria* are predominant in the healthy term infants’ microbiota and are much less represented in preterm neonates and not detected before a post-menstrual age of 30 weeks [27,28].

In preterm infants, a significant amount of the premature microbial gut population is environmentally acquired via the microbial exchange between the room and the occupant. Brooks and colleagues carried out a metagenomic study of the microbes present in 50 preterm newborns and in the environment of the neonatal intensive care unit (NICU) [29]. In both sites, *Staphylococcus epidermidis*, *Enterococcus faecalis*, *Pseudomonas aeruginosa*, and *Klebsiella pneumoniae* were found, and these microorganisms were present in the environment after and often before their detection in the preterm gut. A later study from the same research group suggested that hospitalized preterm infants and their caregivers shaped the microbiomes of NICU rooms [30]. Similar results were presented in a recent multicenter prospective observational study showing that bacterial patterns of VLBW (very low birth weight) newborns at the fourth postnatal week were influenced by NICU practices [31]. Genetic factors may also contribute to the development of the neonatal gut microbiota because it has been found that related twins, although exposed to a different NICU environment, have similar gut microbiota [28].

Soon after birth, the microbial composition is greatly influenced by feeding. Breastfeeding favors the development of a simple community of obligate anaerobes, characterized by higher levels of Bifidobacteria compared to those of formula-fed neonates, which remain elevated after weaning. The low gut bacterial diversity in breastfeeding infants is suggested to depend on the presence of oligosaccharides in human milk, which represent important nutritional substrates for Bifidobacteria [32]. Although *Bacteroides*, *Streptococcus*, and *Lactobacillus* are present in breastfed infants’ microbiota, the microbiota of formula-fed infants is composed of a higher abundance of *E. coli*, *C. difficile*, and *Bacteroides fragilis* [14,33]. Maturation towards an “adult-like” microbiota, mainly composed of Bacteroidetes and Firmicutes, and with increased diversity, seems to depend upon the cessation of breastfeeding, rather than exposure to solid food [34,35]. Such a trajectory towards a mature microbiota, however, does not exclude ongoing adaptation in the microbial community composition during childhood [23,36,37].

## 3. Perinatal Antibiotic Treatment and Neonatal Gut Microbiota

The timing, duration, and type of antibiotic exposure are of particular importance when considering the main factors influencing the composition and function of the neonatal gut microbiota. The impact of antibiotic treatment on the developing neonate’s microbiota may derive both from maternal antibiotic uptake during pregnancy and lactation and from direct exposure to infants, owing to their health status. In the following paragraphs, we consider both conditions, distinguishing between the consequences of direct and indirect antibiotic exposure in term and preterm neonates.

### 3.1. Maternal Antibiotic Exposure

Maternal antibiotic treatment may influence the infant microbiota composition through prenatal exposure to the fetus or changes in the mother’s microbiota or breastfeeding [38]. Intrapartum antibiotic prophylaxis (IAP) is used in up to 40% of women in both elective and emergency cesarean sections and women colonized with group B streptococcus (GBS, *Streptococcus agalactiae*) [39]. Maternal antibiotics may affect neonatal microbial colonization in two ways:-Via the umbilical cord—antibiotics reach the fetus’s blood and they persist up to at least ten hours after administration [25,40].-Antibiotics can modify the maternal vaginal and intestinal microbiome, causing an alteration in vertical microbial transmission and post-natal immunity [25,41].

Antibiotic treatment during labor modifies the development of the gut microbiota in preterm neonates, reduces intestinal host defenses, causes some alterations in the vaginal microbiota before delivery, and influences the composition of the neonatal oral microbiota [42]. Perinatal antibiotic prophylaxis is useful to prevent GBS infection in neonates. Cephazolin and benzylpenicillin are the two most frequently used intrapartum antibiotics which reduce different strains of oral streptococci and this could explain the decreased concentration of the Streptococcaceae family in infants born to mothers who received intrapartum antibiotics. Oral streptococci are important for the establishment of later colonizers due to the production of polysaccharides and adhesins that recruit Gram-positive and Gram-negative bacteria. As a result, the altered oral streptococcal concentration leads to oral colonization by Proteobacteria, which is a cause of dysbiosis and inflammation [42,43].

IAP may also lead to decreased diversity and a lower concentration of *Lactobacilli* in the vaginal microbiota, which may be associated with a high risk of preterm delivery and may cause an increased risk of vaginal GBS infection [44,45,46]. In particular, IAP is associated with a lower relative concentration of *Actinobacteria*, especially *Bifidobacteriaceae*, and a more relative abundance of *Proteobacteria* compared with non-exposed infants [25,39,47]. It is known that *Bifidobacteria* may stimulate genes which promote mucosal integrity and reduce the expression of inflammatory genes, whereas *Proteobacteria* are associated with metabolic and inflammatory diseases [25,48].

Finally, there are few data available regarding the effects of IAP on breast milk composition. It has been reported that mothers with IAP have a lower concentration or absence of *Bifidobacterium* spp. in their milk [39,49,50].

### 3.2. Neonatal Antibiotic Exposure: Consequences for Term Infants

The neonatal gut microbiota is influenced by the timing, the duration, and the type of antibiotic exposure. Empiric antibiotic therapy is associated with lower intestinal bacterial diversity and a major concentration of *Enterobacter* in term infants during the first month [51].

In full-term infants, antibiotic administration during the first hours of life reduced the level of *Bifidobacterium* in the days immediately after birth and subsequently increased the levels of *Enterobacteriaceae* [52] (Figure 1).

It is important to point out that dysbiosis of the gut microbiota is probably an important risk factor for early-onset sepsis (EOS), as shown by Zhou et al., because patients with EOS have a different microbiota composition, such as a high concentration of *Bifidobacterium* and *Staphylococcus* spp. [44].

A retrospective, cross-sectional study led by Rooney et al. disclosed that in all infants within one week of discontinuation of the therapy, each additional day of antibiotics was associated with a lower concentration of obligate anaerobes such as *Bifidobacteria*, *Lactobacilli*, *Bacteroides* and butyrate-producers such as *Bifidobacteriaceae*, *Bacteroidaceae*, *Eubacteriaceae*, *Fusobacteriaceae* at the end of therapy [48,53] (Figure 1). A low Bacteroidetes concentration has been associated with an increased risk of developing type 1 diabetes, asthma, and allergic disease [54].

It has been also demonstrated that neonatal antibiotic therapy may be associated with decreased growth during the first year of life, whereas during infancy and childhood it might be linked to an increased risk of overweight and obesity [55]. This decreased growth is more severe in neonates who receive a full course of antibiotics. The possible cause of this growth impairment may be antibiotic-mediated dysbiosis, such as the reduction of *Bifidobacterium*, which plays an important role in the digestion of dietary compounds and modulates host energy metabolism and satiety [55]. Additionally, impaired childhood growth is associated with poor neurodevelopment outcomes [56,57] and an increase in cardiometabolic risk factors later in life [58,59].

The duration of therapy also increases the risk of *Clostridium difficile* or antimicrobial resistance. For these reasons, co- or post-administration of probiotics may represent a feasible treatment aimed at reducing the effects of antibiotics on gut microbiota [53]. Actually, probiotics may restore the intestinal micro-ecological balance as well as the intestinal barrier and improve intestinal colonization by inhibiting the excessive growth of opportunistic pathogens [60].

### 3.3. Neonatal Antibiotic Exposure: Consequences for Preterm Infants

Antibiotic use is a common neonatal practice for the prevention and treatment of sepsis, which is one of the main causes of mortality and morbidity in preterm infants [38].

Preterm neonates have a different gut microbiota composition, with the delayed colonization of common bacteria such as Bifidobacteria and Bacteroides and an increased concentration of pathogens such as Clostridia (Figure 1). This differing microbiota composition may be due to premature birth, the influence of the hospital environment, medical interventions after birth, and the use of antibiotics [44,61]. Unfortunately, for various reasons, such as chorioamnionitis or the premature rupture of membranes, prolonged antibiotic therapy is frequently applied for these babies and perinatal antibiotic exposure may increase the risk of late-onset sepsis (LOS) and NEC [51,52]. This is probably due to an overall decrease in IL-17A production in the gut and an increase in bacterial translocation [62].

Gibson et al. demonstrated that the administration of meropenem, cefotaxime, and ticarcillin-clavulanate in preterm infants led to a significantly reduced gut microbiota diversity, whereas ampicillin, vancomycin, and gentamicin had non-uniform effects on species richness [63].

In an observational study led by Zwittink et al., short- (≤3 days) and long-lasting (≥5 days) intravenous antibiotic treatment (amoxicillin/ceftazidime) during the first postnatal week in 15 late preterm infants (35 ± 1 weeks of gestation) severely affected their normal intestinal colonization. Both treatments had negative effects on the *Enterobacteriaceae* family, reduced the concentration of *Bifidobacterium*, and increased the presence of *Enterococcus* for up two weeks after the discontinuation of treatment, which represented a health risk for the infants [38] (Figure 1). Only the short-term therapy allowed the recovery of *Bifidobacterium* concentrations within the first six postnatal weeks, which could control other bacterial species and play an important role in early-life tolerance induction and immune system maturation [38]. The use of broad-spectrum antibiotics during the first week of life is suggested to be associated with different levels of inflammatory markers such as sVCAM-1, sCD14, sCD19, sCD27, IL-1RII, sVEGF-R1, and HSP70 (a stress-responsive protein) at 1 year of age [64]. In addition, infantile colic during the first 3 months of life is associated with increased inflammatory markers such as IL-33, whereas children with eczema have a reduced capacity to induce Th1 cytokines (such as IFN-γ and CXCL9) [64].

Other studies confirmed that a short-term antibiotic treatment lasting less than three days in infants may have only mild and temporary effects on the gut microbiota composition and their metabolites [52,65].

Zhu et al. analyzed the stool microbiota and metabolites in 36 preterm neonates divided into three groups. Two were treated with penicillin and moxalactam or piperacillin-tazobactam for 7 days, whereas one was antibiotic-free. Both treated groups exhibited a reduction of gut bacterial diversity and an increase in dangerous bacteria such as *Streptococcus*, which can cause serious infections such as neonatal sepsis, and *Pseudomonas*. The piperacillin-tazobactam group also exhibited an overgrowth of *Enterococcus*, which is intrinsically resistant to different antibiotics and may cause nosocomial infections [51].

In this way, probiotics may help to reduce the incidence of necrotizing enterocolitis (NEC), late-onset sepsis, and the mortality of preterm neonates [60].

Early-life antibiotic therapy could increase the presence of pathogenic, antibiotic-resistant *Enterobacteriaceae*, which are especially resistant to beta-lactam antibiotics [66]. Additionally, in the genomic context, the expression of antibiotic resistance genes may be influenced by exposure to antibiotics in early life [67]. There are several pieces of evidence related to the development of multidrug resistance genes in Gram-negative bacteria, especially after prolonged antibiotic treatment. Third-generation Cephalosporins or Carbapenems are more frequently associated with the development of antibiotic resistance among Gram-negative bacteria than Aminoglycosides [48,68,69,70]. Antibiotics also select bacteria expressing resistance genes in the gut microbiota [48].

Trasand et al. demonstrated that antibiotic exposure during the first six months of life was associated with increases in body mass from 10 to 38 months, whereas later exposure (6–14 months, 15–23 months) was not consistently associated with an increased body mass [71]. Subtherapeutic antibiotic doses may be associated with an increased body mass in farm animals and laboratory mice, whereas this has not been observed in children with repeated antibiotic treatment [67]. In a systematic review, the antibiotic-induced reduction of gut microbiota diversity was associated with various long-lasting consequences, such as obesity and inflammatory disease [48,72]. Perinatal exposure to broad-spectrum antibiotics such as ampicillin induced dysbiosis, which seemed to be associated with alterations of colonic CD4+ T cells and in particular of neuropilin-negative RORyt+ and Foxp-3 positive Tregs. This change may lead to compromised immune tolerance, with the development of immunologic and metabolic disease [72].

Finally, in a systematic review that identified 129 studies, alterations of human intestinal microbiota due to antibiotics exposure were reported. In particular, amoxicillin, amoxicillin/clavulanate, cephalosporins, macrolides, clindamycin, tigecycline, quinolones, and fosfomycin increased the abundance of *Enterobacteriaceae* other than *E. coli* (especially *Citrobacter* spp., *Enterobacter* spp., and *Klebsiella* spp.). Amoxicillin, cephalosporins, macrolides, clindamycin, quinolones, and sulphonamides decreased the concentration of *E. coli*, whereas amoxicillin/clavulanate, in contrast to other penicillins, increased the abundance of *E. coli*. Piperacillin and ticarcillin, carbapenems, macrolides, clindamycin, and quinolones strongly decreased the abundance of anaerobic bacteria [73].

## 4. Possible Strategies for the Prevention of Dysbiosis and Restoration of the Microbial Community after Antibiotic Exposure

In the literature, different pieces of evidence are emerging related to microbiome-based approaches for the restoration of a healthy microbiota in neonates.

Antibiotic treatment in neonatal infections has important negative effects on gut microbiota due to the reduction of the populations of *Bifidobacterium* and *Lactobacillus*. These bacterial species may promote gut health and may prevent pathogen colonization [74]. Although the use of probiotics such as *Bifidobacterium longum*, *Lactobacillus acidophilus*, and *Enterococcus faecalis* seems not to restore the composition and diversity of the neonatal gut microbiota, the simultaneous use of probiotics and antibiotics might have a more beneficial effect by promoting an increase in the presence of *Bifidobacterium* [75].

Single probiotic strains, such as *Saccharomyces bourlardii*, *Lactobacillus reuteri*, *Lactobacillus acidophilus* and *Bifidobacterium lactis*, have shown modest improvements in gut health, whereas the greatest success has been observed in probiotic mixes [74]. The use of three to eight different bacterial strains may reduce LOS in enterally-fed infants because different bacterial species may have complementary roles in restoring gut health and providing protection [76,77]. However, probiotics with a single strain or a mixture of *Lactobacillus rhamnosus GG*, *S. boulardii*, *L. reuteri*, *Lactobacillus sporogenes*, or *B. breve* have shown limited effects in reducing LOS in formula-fed preterm neonates [78]. These data have suggested that nutrition might influence the efficacy of probiotics.

In a 10-year observational study led by Beck and colleagues, it was demonstrated that supplementation with two different probiotics in preterm infants—one with *Bifidobacterium bifidum* plus *L. acidophilus* and the second one with *B. bifidum*, *B. longum* subsp. *infantis* and *L. acidophilus*—was associated with a faster transition into two different *Bifidobacterium* spp. and to positive health outcomes [79]. In these neonates, an important bifidogenic role was performed by breastmilk [79]. Breastmilk-fed neonates have a higher concentration of *Bifidobacterium* and *Bacteroides*, whereas formula-fed infants maintain *Enterobacteriaceae* for longer periods of time [80]. Human milk oligosaccharides (HMOs), which are multifunctional glycans that are naturally present in human milk [81], counteract the negative effects of antibiotics by increasing the levels of *Bifidobacteria* and decreasing those of staphylococci [79] (Figure 2). *Bifidobacterium* produces short-chain fatty acids (SCFAs) that reduce gut permeability and preserve the integrity of the intestinal barrier [82]. In addition, HMOs seem not only to have prebiotic effects on commensal bacteria but also to exhibit antimicrobial activity against pathogens such as group B *Streptococcus* (GBS), which is the main organism responsible for neonatal infections [83] (Figure 2). HMOs may serve as a substrate to modify the growth of GBS. They also have anti-adhesive effects because they work as soluble ligand analogs and block pathogen adhesion [81]. Finally, they seem to have glycomic modifying effects by altering glycan expression on the epithelial cells, reducing bacterial attachment [81]. HMOs also provide protection against viral infections because they promote the maturation of the immune system, along with a more balanced Th1/Th2 cytokine response, and stimulate commensal bacterial growth [81].

In this way, breastmilk components, such as HMOs, growth factors, immunological factors, and probiotic bacteria could establish a healthy gut microbiota and might promote more effective crosstalk among probiotics, gut microbiota, and the immune system [78].

Some evidence has revealed that supplementation with *Bifidobaterium*-containing probiotics during hospitalization after preterm birth might prevent the persistence of antibiotic resistance genes in the gut microbiome, such as aminoglycoside and beta-lactam resistance [84,85,86].

Finally, probiotic, prebiotic, or synbiotic administration to cesarean-born neonates might be associated with improvements in some health outcomes through their immunomodulatory effects, such as decreasing atopic and infectious diseases or increasing the immune vaccination response [87].

## 5. Conclusions

During prenatal and postnatal life, infants are colonized by different microorganisms. Gut colonization is a dynamic process, influenced by a wide range of factors such as the mode of delivery, diet, environment, feeding, and antibiotic treatment. Antibiotic-induced alterations of the gut microbiota might affect the maturation of the immune system and increase susceptibility to obesity, diabetes, inflammatory bowel disease, and immune-related diseases such as asthma and allergies. Additionally, dysbiosis causes an increased risk of early adverse outcomes such as NEC, sepsis, and fungal infections. However, the rapid interruption of antibiotic treatment and the use of probiotics may allow for the recovery of the composition of the gut microbiota and appear to have beneficial effects on human health.

An important possible future strategy could involve the consideration of microbiota as a possible intervention target to promote infants’ health, because full-term delivery, a lack of perinatal antibiotic exposure, and the use of probiotics seem to be protective factors against gut dysbiosis and all its consequences.

## Figures and Tables

**Figure 1 antibiotics-12-00258-f001:**
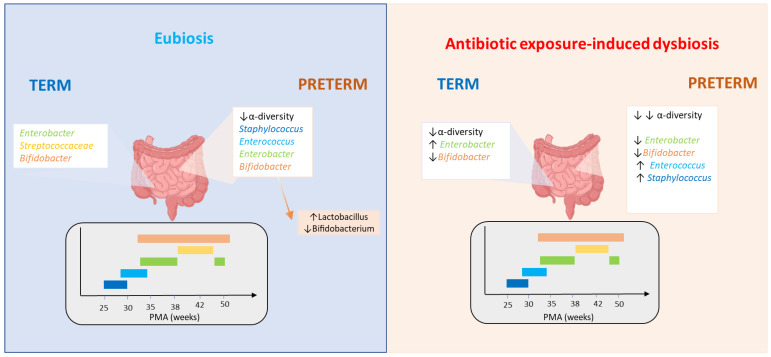
Effects of antibiotic therapy on neonatal gut microbiota according to postmenstrual age (PMA). In the blue box (eubiosis conditions) the colors in the graph indicate the expression of the relevant bacterial species in term and preterm infants according to PMA. The pink box represents the bacterial changes induced by antibiotic exposure in term and preterm infants according to PMA.

**Figure 2 antibiotics-12-00258-f002:**
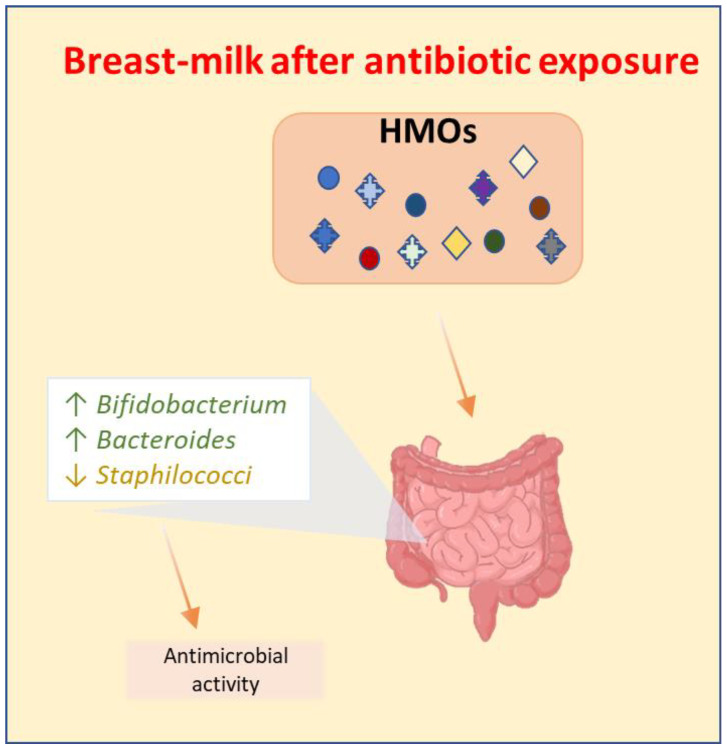
Beneficial effects of breast milk on gut microbiota after antibiotic exposure. Human milk oligosaccharides (HMOs) increase the intestinal concentration of *Bifidobacterium* and *Bacteroides*, decrease *Staphilococci*, and have also exhibited antimicrobial activity.

## Data Availability

Not applicable.

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
