# Peer review of "Effects of Perinatal Antibiotic Exposure and Neonatal Gut Microbiota"

_antibiotics, 2023, doi:10.3390/antibiotics12020258_

Round 1
Reviewer 1 Report
In this review, Morreale et al discuss the negative consequences of prenatal antibiotic treatments on the neonatal health by disrupting the gut microbiome imbalance. The review is really well researched and written. It explains in detail how in-utero exposure to mother’s metabolites and antibiotics may influence fetal development, along with briefly suggesting that alternative treatments like probiotics may restore the microbial composition in the newborns.
Author Response
Thank you for your feedback.
Reviewer 2 Report
Good work. Something must be clarified:
Line 249 and 250 - Figure 1
11. Explain abbreviation PMA
22. The title of the figure 1 is: ,, Figure 1. Effects of antibiotics therapy on neonatal gut microbiota according to gestational age”.
The duration of the gestation is 40 weeks, maximum 42 weeks. Explain the gestation age of 50 weeks. This is not possible. If 50 weeks is not referred to gestational weeks, explain the meaning of this number. If it is referred postnatal period, you should precise this on graph. I think that you should change this figure to clarify the exact meaning.
This figure needs better explanation in subtitle or in the text of the article.
Author Response
Thank you for your suggestions.
11. I have explained the meaning of PMA (post-mentrual age)
22. I have added a better explanation of the Figure 1 in the caption. 50 weeks means 50 weeks of PMA
Reviewer 3 Report
Dear authors
Thank you for giving us the opportunity to review the manuscript. This review systematically describes the effects of antibiotic treatment on neonatal intestinal flora and subsequent negative effects on infant health. The article is well-structured, well-written and comprehensive. After careful reading and review of “antibiotics-2160620”, we propose the following review comments and suggestions.
There are few contents about optimizing antibiotic therapy and regulating gastrointestinal flora by probiotics, so it is suggested to add some contents after systematic search. We recommend receiving after minor revisions.
Yours
Chen
Author Response
Thank you for your suggestions.
I have added other informations in the last paragraph especially about the effects of probiotics.
We hope it is satisfactory
Reviewer 4 Report
Congratulations to the authors for a good and valuable work. You draw attention to a very important issue related to the danger of disturbing the homeostasis of intestinal microbiota.
However, I have a few comments:
In my opinion, the title of paragraph 4 should be changed. The current name is "Alternative treatments to antibiotic therapy and approaches aimed at restoring the microbial community". I think that in this part the authors do not present alternatives to antibiotic therapy, but rather emphasize the role of prevention, so this title is inadequate to the content.
I would include more information on the role of breast milk in shaping the baby's intestinal microbiota. The authors write about it, but it is so important that it is worth develop.
In section 3.2. "Neonatal antibiotic exposure: consequences on term infants" in the first sentence, the authors refer to the situation of preterm infants. Another paragraph is dedicated to them, so this information is not needed here.
Author Response
Thank you for your suggestions.
- I have changed the title of the last paragraph in "Possible strategies of dysbiosis’ prevention and restoration of the microbial community after antibiotic exposure"
- I have added some new informations about the effects of breat milk in the last paragraph and a Figure to better explain what is written in the text (Figure 2).
- I have shifted the sentences about the preterm in the right paragraph.